# The Power of Reagent Titration in Flow Cytometry

**DOI:** 10.3390/cells13201677

**Published:** 2024-10-11

**Authors:** Diana L. Bonilla, Alberta Paul, Jesus Gil-Pulido, Lily M. Park, Maria C. Jaimes

**Affiliations:** 1Scientific Commercialization, Cytek Biosciences, Inc., 47215 Lakeview Boulevard, Fremont, CA 94538, USA; 2Customer Education, Cytek Biosciences, Inc., 47215 Lakeview Boulevard, Fremont, CA 94538, USA

**Keywords:** flow cytometry, antibody titration, spectral cytometry, data reproducibility

## Abstract

Flow cytometry facilitates the detection of multiple cell parameters simultaneously with a high level of resolution and throughput, enabling in-depth immunological evaluations. High data resolution in flow cytometry depends on multiple factors, including the concentration of reagents used in the staining protocol, and reagent validation and titration should be the first step in any assay optimization. Titration is the process of finding the concentration of the reagent that best resolves a positive signal from the background, with the saturation of all binding sites, and minimal antibody excess. The titration process involves the evaluation of serial reagent dilutions in cells expressing the antigen target for the tested antibody. The concentration of antibody that provides the highest signal to noise ratio is calculated by plotting the percentage of positive cells and the intensity of the fluorescence of the stained cells with respect to the negative events, in a concentration–response curve. The determination of the optimal antibody concentration is necessary to ensure reliable and reproducible results and is required for each sample type, reagent clone and lot, as well as the methods used for cell collection, staining, and storage conditions. If the antibody dilution is too low, the signal will be too weak to be accurately determined, leading to suboptimal data resolution, high variability across measurements, and the underestimation of the frequency of cells expressing a specific marker. The use of excess antibodies could lead to non-specific binding, reagent misuse, and detector overloading with the signal off scale and higher spillover spreading. In this publication, we summarized the titration fundamentals and best practices, and evaluated the impact of using a different instrument, sample, staining, acquisition, and analysis conditions in the selection of the optimal titer and population resolution.

## 1. Introduction

Antibodies (Ab), also called immunoglobulins (Igs), are glycoproteins produced by B cells, which bind to specific antigens. Upon binding, immune recognition starts, with the subsequent target identification, clearance, or neutralization [1]. Each Ab is formed of four polypeptides, with two heavy and two light chains, joined together by disulfide bonds, and forming a Y-shaped molecule. Each chain has constant and variable regions, being the variable ones critical for specific antigen binding. Igs can be divided into five subclasses (IgA, IgE, IgG, IgD, and IgM), according to their structure, number of binding sites, and functionality. Igs can be enzymatically cleaved into two distinct portions: the antigen binding fragment (F(ab’)2) with antigen-binding activity, and the crystallizable fragment (Fc) which directs the biological activity of the molecule, by binding to complement or Fc receptors [2]. 

Monoclonal antibodies (mAb) specifically bind to a unique epitope in the antigen (Ag), together forming an immune complex [3]. Antibody binding depends on the concentration and configuration of both parts of the complex, the dynamics of association and disassociation of the reaction, the dissociation constant, the reversibility of the binding, and whether an equilibrium in the reaction is reached. Affinity is the strength of the immune complex binding, as Abs can bind to high or low affinity targets. Avidity is the overall strength, impacted by factors such as affinity, the arrangement of the binding parts, the number of binding sites in the Ab, and the multivalency of the Ag. For example, IgM has low affinity but high avidity due to its 10 Ag binding sites. Other factors that can influence binding include molecule orientation; changes in protein folding, charge, or glycosylation [4]; the reversibility of the binding based on type of bonds and temperature; or target accessibility. mAbs are monospecific in nature, with affinity and specificity for only one epitope of a selected antigen. They can be commercially produced by cloning B cells fused with tumor cells in a hybridoma or using more recent technologies such as phage display [5]. The use of mAbs is widespread in research and clinical laboratories, and in a variety of analytical assays, including immunohistochemistry, immunoblotting, enzyme-linked immunosorbent assays, and flow cytometry. 

The use of any mAb for analytical assays requires validation to confirm that the reagent binds only to the intended target. Multiple publications have addressed the challenges when using mAbs as part of the reproducibility crisis in scientific communications [6,7,8]. Issues with cross-reactivity, lot-to-lot variability, lack of specificity or reproducibility, and scarcity of reference materials have led to inaccurate conclusions and the misspending of time and resources. For example, Michel et al. reported a lack of sensitivity of 49 antibodies commercially available to identify G-protein-coupled receptors. The authors emphasized that antibodies should be validated by the disappearance of staining in the absence of the target, selectivity of staining compared to other targets, and similar staining patterns to other antibodies raised against different epitopes in the same Ag [9]. Kalina et al. showed differences in reactivity for two CD85d clones. Clone 287219 recognized other CD85 family members (a, b, and c), while clone 42D1 had no cross reactivity [10]. Moreover, different fractions or batches of the same Ab can have different binding strengths or degrees of labeling, which could be mitigated by standardizing the conditions under which the reagents are produced. 

In 2016, a proposal was signed by more than one hundred investigators to emphasize the need of guidelines for Ab use and validation, and since then, many portals and working groups have documented how to identify successful antibodies and improve data reproducibility [11]. The antibodypedia website has catalogued and rated more than 4 million commercially available antibodies, based on validation data available (www.antibodypedia.com accessed on 5 March 2023), and the Resource Identification Portal (RRID) has a repository of high-quality reagents and guidelines for consistency and transparency in the use of antibodies in scientific publications (https://scicrunch.org/resources accessed on 5 March 2023). In summary, a high-quality mAb should specifically recognize its target with minimal off target binding, and the signal should disappear if the target is removed when using the same analytical method or a different confirmatory assay.

Flow cytometry uses fluorochrome-conjugated antibodies to identify the presence of proteins expressed by cells or small particles [12,13]. Figure 1 describes the process of fluorescence excitation and emission using a spectral flow cytometer, based on its optical configuration. Many publications have detailed best practices for antibody titration as part of flow cytometry panel validation and evaluation [7,14,15,16,17,18]. Differences in reagent performance can be observed due to Ab aggregation, the type of fluorochrome conjugated to the Ab, conjugate stability, fluorochrome interactions, Ab interactions, fluorochrome size or charge, protein-to-fluorochrome ratio, and the steps and reagents used in the staining protocol [19]. For example, it has been shown that CD25 clone BC96 is more sensitive to fixation, a step commonly used in immunostaining, compared to clones 4E3 or M-A251. Doyle et al. reported that some Ab clones are sensitive to tissue digestion [20]. In cells expressing Fc receptors, the Ab can function as a ligand and bind the receptors through the Fc portion. Andersen et al. showed the effectiveness of Fc receptor blocking agents, prior Ab staining, to reduce the staining background in monocytes [21]. It is critical to confirm that every Ab works using the staining protocol and cell type intended to be used in the actual experiment. For example, Park et al. described the titration process followed for every single fluorescent reagent when optimizing a 40-color spectral cytometry panel [13,22]. A titration protocol was included to improve resolution for a highly multiparametric assay; data analysis and titer modifications were needed due to steric hindrance and were provided. In a similar way, Fernandez et al. also reported some protocol modifications needed to prevent steric hindrance, with a nice example of Ab binding to a recombinant IgG reagent [23]. 

In this manuscript, the impact on marker resolution and the titration performance of sample preparation protocols, staining conditions, instrument designs, software features, and data analysis strategies was evaluated, and guidelines are provided for the selection of the optimal titer and population resolution.

## 2. Methods

### 2.1. Materials 

Flow Staining Buffer (1×) (Tonbo, catalog TNB-4222-L500);1× Phosphate-Buffered Saline (PBS);V-bottom 96-well plates;Multichannel pipette with a range 15–300 µL;Centrifuge with plate adapters;Paper towels.

### 2.2. Antibody Dilution Preparation 

Determine the antibody stock concentration by referring to the antibody product sheet or certificate of analysis (CoA). Antibodies typically come with use recommendations in μL/test or μg/mL. NOTE: To find the Ab stock concentration when using cFluor^®^ reagents, use this link to the CoA finder: https://cytekbio.com/pages/certificate-of-analysis-locator accessed on 30 March 2023.Calculate the antibody and buffer volume required for the first dilution.For antibodies which come as mg/mL, start dilutions at 1000 ng/test, in a final volume of 200 μL, or adjust based on volume used for the multicolor staining.For antibodies that come as μL/test, start dilutions at double the recommended volume in a final volume of 300 μL.Prepare the first dilution for each antibody to be titrated in its designated well. We recommend an 8–12-point titration in 96-well plates.Add 150 μL of stain buffer to the remaining wells.Perform 2-fold serial dilutions. Using a multichannel pipette, set to 150 μL, mix the antibody and buffer in the first column and transfer half the volume (150 μL) to the second column. Using the same tips, mix 5 times before transferring 150 μL to the next column.Repeat for all wells and remove 150 μL from the last well.Store in the dark until the cells are ready.

### 2.3. Cell Preparation 

Use PBMCs. Resuspend the cells in staining buffer at 2 × 10^6^ cells/mL. Make sure you have enough cells for all titration wells and use the same cell number for each well. NOTE: If titrating antibodies for rare or low expressing markers, you may need to increase the total number of cells being stained by increasing the initial cell concentration or acquire more events. You might need to add other reagents depending on your assay optimization, including Monoblock, to prevent monocyte binding, Cellblox in case of using Novafluors, or Fc block to prevent Fc binding.Add 100 μL of sample to your titration wells, such that the final volume is 250 μL. This volume must match the volume that will be used in the experimental samples.Pipette up and down to mix, avoiding the formation of bubbles.Incubate for 20 min at room temperature in the dark, or according to your specific staining protocol.Centrifuge 5 min at 400× *g*, decant supernatant, and blot on paper towels.Resuspend in 200 μL of staining buffer.Repeat steps 6 and 7 twice.Store the plate at 4 °C in the dark until the samples are analyzed. NOTE: If any details differ from your specific staining conditions, follow the steps (temperature, incubation time, washes, fixation, permeabilization) of your own protocol.

### 2.4. Sample Acquisition 

Warm up your cytometer and verify that it is clean and passing QC.Use optimized acquisition settings for fluorescence detectors. Adjust FSC, SSC, and SSC-B gains to place the cell populations on scale. Adjust FSC threshold to exclude debris.Acquire the samples in plate mode if a loader is present. Alternatively, transfer samples to tubes for acquisition. Ensure enough events are recorded for each sample to clearly identify the positive populations. NOTE: For rare or low-density antigens, you may need to collect more cells than for most lineage markers.

### 2.5. Data Analysis

We recommend the calculation of stain index and percentage of positive cells.Perform data cleanup by gating the population in which your antigen should be expressed using a FSC vs. SSC plot. Exclude doublets via a single cell gate using FSC-H vs. FSC-A.Verify fluorescence signal stability by checking the peak detector vs. time. You should observe consistent signals across time.Create a histogram for the peak detector for the fluorochrome being titrated and set a gate on the negative and positive peaks. You may need to adjust the gates between each sample dilution iteration.Confirm that the positive peak has the expected full spectrum profile for the fluorochrome being tested, and that the negative peak does not contain any unexpected signatures.Create a statistics table with the % of positive and negative events in the parent gate, and the median (MFI) and robust standard deviation (rSD) for the positive and negative populations.Export the statistics table and calculate the stain index for each dilution. The stain index is the difference between the positive and negative medians divided by two times the robust standard deviation of the negative population (MFI positive-MFI negative)/(2 × rSD negative).Plot the stain index value against the Ab concentration for each dilution.Concatenate the fcs files to help visually compare the dilutions.Inspect the plot for the optimal concentration range. Low concentrations result in insufficient staining and a lower stain index. Saturating concentrations are reached when all Ag binding sites are bound by Ab, without excess unbound Ab, and minimal non-specific binding. At saturating concentrations, the stain index is at its peak and plateaus. Concentrations over the saturation phase may decrease the stain index due to higher background or Ab aggregation.Plot the percentage of positive cells against Ab concentration and ensure the chosen antibody concentration captures all positive cells.

### 2.6. Calculation of Antibody Optimal Titer 

Optimal Ab concentration is a function of the relationship between the number of antibodies and the number of antigen binding sites. The optimal titer is defined by the Ab concentration that best resolves the positive signal at saturation.The optimal titer lies on the plateau area, and it can be defined with statistical criteria using a non-linear regression analysis to model the data. We use the stain index as the readout.In general, the concentration of reagent to use should be at least 2-fold above a concentration, which gives at least 90% of the saturating staining.

### 2.7. Experimental Design

Human peripheral blood mononuclear cells (PBMCs) or fresh mouse spleen cells were used. Titers were reported as ng/test, with a test being equivalent to the final staining volume per tube. Most samples were acquired in the spectral cytometer Cytek^®^ Aurora (Cytek Biosciences, Fremont, CA, USA), which was equipped with 5 lasers and 64 fluorescence detectors. The instrument was used at the settings recommended by the manufacturer. For the spectral cytometer, unmixing was performed in SpectroFlo software. Data were analyzed using SpectroFlo version 3.1.0 and FCS Express 7.0.

## 3. Results

Flow cytometry titration data are usually displayed as concatenated images of all concentrations assessed. Figure 2 illustrates a titration example of a human anti-CD4 Ab (clone SK3) conjugated to the cFluor BYG781 fluorochrome. Panel A shows a scheme on how the twelve two-fold sequential Ab dilutions were prepared in a 96-well plate. Panel B shows a concatenated image of the serial Ab dilutions tested in human lymphocytes. The intensity of the median fluorescent signal (MFI) is displayed on the Y-axis and the increasing Ab concentrations are displayed on the X-axis, with titers ranging from 0.5 to 1000 ng in a staining volume of 200 µL for 100,000 cells per test. The increase in BYG781 fluorescence intensity is observed in a dose–response manner in the positive population, plateauing around 32 ng/test. All the dilutions are separating negative and positive events, but only saturating titers provide maximal separation. An increase in background in the negative population is also observed, especially at higher concentrations. 

Panel C dissects the titration curve in detail. At low non-saturating antibody concentrations (light grey area), maximum fluorescence intensity is not achieved, resulting in insufficient staining and the lowest separation. Since the saturation point is not reached, minimal changes in the staining protocol, such as the temperature of the reaction, incubation time, washes, or even pipetting errors, will have a massive impact on the resolution with a higher sample-to-sample variation. The positive population gradually increases in intensity and rises to a plateau when the reaction reaches equilibrium due to Ab saturation. The optimal titer lies in this portion of the curve (green section), where the positive signal has its highest intensity and plateaus with a minimal background. The effect of staining variables is minimal in the saturation phase. Most Abs for flow cytometry saturate below 1 μg per test; however, there are some exceptions where Abs saturate at exceedingly high concentrations or at extremely low concentrations. The final portion of this curve shows oversaturating concentrations where the excess of Ab leads to a higher background without improvement in separation (dark grey area). Representative SSC vs. CD4 pseudo color plots are shown to illustrate the separation on each phase. It is noteworthy that in some cases, a lower intensity can be observed at higher concentrations due to Ab aggregation that prevents binding, a prozone effect where less immune complexes are formed under excess of Ab, fluorochrome quenching, or more monovalent binding forced at higher concentrations caused by Ag binding sites saturating faster. 

The titration goal is to identify the dilution with the best separation between positive and negative events. Figure 3A shows a histogram with both positive and negative events using the CD4 cFluor BYG781 example. Based on this staining pattern, cells can be gated and evaluated for their intensity of fluorescence, spread of the data, and frequency. Multiple resolution metrics (signal to noise ratio, stain index, separation index) as well as the percentage of positive peaks can be used to evaluate the optimal titer (Figure 3B). Michaelis–Menten kinetics can be applied to Ab-Ag binding, assuming the reaction follows a single substrate enzymatic dynamic. The titration curves show the plot Ab concentration (substrate) against each metric, representing the product rate or velocity of reaction (V). At low substrate concentrations, the velocity is linearly proportional to the concentration, but at higher substrate concentrations, the rate of change in velocity as a function of the substrate concentration will be negligible, due to the saturation of binding sites. Vmax is the maximum rate achieved by the reaction. 

A linear regression can be used to fit the curve where the optimal Ab concentration to use should be above 90% of Vmax, assuming that saturation is the point where more than 90% of the epitopes are forming immune complexes with Ab. In this example, the optimal concentration is set at 32 ng per test. Concentrations over 32 ng do not improve the separation, due to the higher background. If the concentrations evaluated are non-saturating or if the saturation is reached in the lowest titers, the titration needs to be repeated by adding concentrations at a higher or lower range. For example, when the CD4 BYG781 reagent was first assessed between 8 and 1000 ng per test, the progression of the curves at the lower end could not be evaluated and additional dilutions below 8 ng were included to determine the behavior before the saturation point.

We analyzed data from more than 200 titrations, with a variety of specificities and fluorochrome conjugates, to evaluate the impact of multiple factors in the Ab titer: resolution metrics, cytometer types, wavelengths, acquisition settings, sample preparation, data file types, staining conditions, and autofluorescence extraction strategies. 

### 3.1. Impact of Metrics Used for Optimal Titer Calculation

The optimal Ab point is defined by the concentration that resolves a better positive signal from the background. We compared multiple metrics (signal to noise ratio or S/N ratio, stain index, and separation index) to determine which one is more informative. The titration of human CD4 APC (clone SK3) shows the effect of increasing Ab concentration on the signal intensity. In the concatenated plot, we observed gradual increments of the signal plateauing around 32 ng/test, as shown before for this marker (Figure 4A). All dilutions evaluated (between 0.5 and 1000 ng/test) were separating the positive populations, while only titers over 16 ng reflect saturating concentrations. The graph shows a comparison of the stain index, separation index, S/N ratio, and percentage of positive cells (secondary Y-axis), in an Ab concentration-dependent manner (X-axis). All calculations lead to the same conclusion regarding the optimal titer and curve shape. The percentage of cells expressing CD4 was not perturbed by Ab dilutions (Figure 4B).

Other conjugates for the same specificity and clone (CD4, SK3) were evaluated in parallel to the APC conjugate in PBMCs. Figure 4C shows the concatenated titration of PECy7, PE, and APC conjugates, testing the same clone and Ab concentrations. All the plots show similar titration curve shapes and optimal titer selections. As expected, the fluorochrome brightness impacts the level of separation between positives and negatives. PE, the brighter fluorochrome, leads to higher resolution, followed by PECy7, and lastly, APC. When comparing the resolution metrics across the three conjugates, there were no differences in the curve pattern and optimal titer selection. The percentage of positive cells remained unchanged as well. 

CD4 is expressed at relatively high levels, so an important consideration is if similar conclusions can be made when testing a dimly expressed antigen. Figure 4D,E show the metric comparison for a low expressed marker, TCRγδ, conjugated to PerCP-eVio 700. At low Ab concentrations, the positive populations are barely separated, with saturation at the highest titer. However, the background was detrimentally expanded at 1000 ng with a loss of resolution. All metrics calculated show a similar trend in titration behavior, with the optimal titer defined at 500 ng. The use of the percentage of positive cells was critical for this low expressed marker. MAb concentrations below 125 ng do not capture the entire TCRγδ positive population. This example clearly illustrates that the identification of the optimal titer for antigens expressed at low levels should be made based on both the resolution and percentage of positivity. 

Since a high correlation was observed between resolution metrics, for the rest of this publication, we exclusively used the stain index and percentage of positives as readouts of our titration experiments. Similar observations across metrics were made using other conjugates across the full range of specificities and emissions (380–810 nm), with primary UV (CD45RA BUV395 and CD8 BUV805), Violet (CD3 BV510 and CD20 Pacific Orange), Blue (CD14 SB550 and CD57 FITC (Fluorescein isothiocyanate)), Yellow Green (CD4 CF568 and HLADR PE/Fire810), or Red excitation (CD19 SparkNIR685 and CD27APC-H7).

### 3.2. Impact of Instrument Setup

The optical detector setup used for sample acquisition highly influences population resolution. We assessed whether non-optimal detector settings could impact the resolution and titer definition. Ab dilutions were acquired at the optimized manufacturer-recommended settings (Cytek Assay Setting-CAS), and then those settings were increased or decreased by 50%. The CD4 PerCP-eFluor 710 titration at different settings is shown in Figure 5A. We found that the optimal titer, 125 ng, was not affected by changes in acquisition settings. A higher resolution was observed at both CAS and +50% CAS, while lowering the gains caused a reduction in separation, with the stain index values decreased by 50%. The concatenated images show the reduced separation for CD4 positive events with lower settings. We did not observe differences in percentage of positives.

To evaluate the effect of the instrument setup in the titration of dimly expressed markers, PBMCs were stained CD314 BUV615, with CD8 BB515 as a counterstain, to more accurately identify the double positive cells (Figure 5B). As for the CD4 results, no differences in optimal titer determination (1000 ng/test) were found at different settings, and as expected, a lower stain index was obtained when gains were reduced. However, and importantly, the percentage of positive cells was negatively impacted by using low titers at lower instrument settings, highlighting the importance of using an optimized setup and both the stain index and % positivity for the titration evaluation. The concatenated images clearly show the impact of reducing detector settings in the population separation. The CD314 Ab does not reach saturation at the tested concentrations (no plateau is observed even at 1 µg). Human TCRγδ PE/Fire 700, co-stained with CD3 BV570, and evaluated at different acquisition settings, shows a similar titration pattern across settings with no variation in the titer selection (1000 ng) and a lack of saturation phase (Figure 5C). The same critical role was established for the use of the percentage of positivity in titration evaluation. Comparable results were obtained when testing other specificities. In conclusion, the use of optimized settings provides high resolution, and the optimal titer selection is not perturbed by changes in acquisition settings within the evaluated range.

### 3.3. Impact of Using Raw vs. Unmixed Data in Spectral Cytometers

Data acquisition in spectral cytometers generates what is called raw fcs files. Raw files go through a mathematical process called unmixing to identify the abundance of each fluorochrome in the multicolor sample, using single-stained reference controls. If unmixing is calculated, raw and unmixed files are available for each experiment. We evaluated the effect of using one data file type over the other on the population resolution and optimal titer determination. For raw files, the intensity from the peak detector for each fluorochrome was used for calculations. Figure 6 displays titration results for three markers (CD14 Spark Blue 550, CD57 FITC, and HLA-DR PE/Fire 810). In conclusion, no differences were observed in resolution or optimal titer when using raw vs. unmixed files. Similar observations for the type of fcs files were made using other conjugates across the full range of specificities and emissions.

### 3.4. Impact of Autofluorescence Extraction 

Intracellular components emit fluorescence when excited by the lasers in the cytometer and the fluorescent light can be collected by the detectors in the system. Cellular autofluorescence (AF) can be different depending on the cell type, cell functionality, cell viability, cell stimulation, and even staining conditions. AF impacts the marker resolution by increasing the background in the negative population. To improve resolution when using spectral cytometers, AF can be characterized by acquiring unstained cells to understand the background, and then extracted, by including the AF emission as part of the unmixing calculation, as shown by Kharraz Y et al. [28]. We compared the impact of extracting AF on the resolution and Ab titer definition. AF extraction was performed using SpectroFlo software, and results were compared with files where no AF extraction was applied. 

Unstained lymphocytes and monocytes have differences in AF brightness, with a similar emission spectrum. The AF background impacts the resolution of fluorochromes, such as BUV496 or BUV563, which are emitted in a closer range to the AF profile. For example, in Figure 7A, we observed higher CD16 BUV496 stain indices when AF was extracted, both in lymphocytes and monocytes. However, the optimal titer was not impacted by the extraction (32 ng/test for monocytes, and 500 ng/test for lymphocytes). This titration also illustrates the importance of using the matching cell type, according to the populations present in the experimental samples. Similar conclusions were drawn when testing CCR5 BUV563, with a higher resolution observed upon AF extraction, without changes in titer determination. On the other hand, the resolution of fluorochromes that are emitted in a wavelength range far from the AF emission (CD27 APC-H7, CD19 Spark NIR 585), remain unperturbed (Figure 7B). In conclusion, the identical optimal titer is identified in both AF-extracted and non-extracted samples. As expected, AF extraction improves the resolution of AF-emitting dyes. 

### 3.5. Impact of Sample Preparation or Staining Conditions

The technical conditions used during the sample preparation can modify the Ab binding and titration results, and the impact of the sample preparation depends on the specific marker of interest. Figure 8A shows the Ab titer comparison in whole blood and PBMCs for CD45RA Alexa Fluor 488 and CD3 Spark Blue 550. For CD45RA, there were no differences in resolution, titration curve shape, or optimal titer. In contrast, a higher resolution was achieved for CD3 when using whole blood. These examples illustrate the need to perform titrations with cells prepared in the same way as the experimental samples. 

Figure 8B shows single titration plots for CD57 cFluor B548, comparing human PBMCs and whole blood. The resolution, percentage of positives, and optimal titer selection are influenced by how cells have been processed, with a lower positivity and separation observed in whole blood. Optimal titer differences are shown in red boxes. These observations agree with the publication by Maciorowski et al. [15], emphasizing that careful titration is needed under the same staining conditions as the multicolor samples to preserve the resolution for every marker. Also, it is worth highlighting that differences in the sample type will have an impact as well. If CD16 is tittered in PBMCs, most of the Ab will bind to monocytes or Natural Killer (NK) cells, however, if it is tittered in whole blood, the granulocytes will bind the majority of the Ab, while other target cells will be under tittered.

### 3.6. Impact of the Flow Cytometry Instrumentation

The optimal antibody concentration is a function of cell biology and based on the relationship between the Ab concentration and the number of Ag binding sites. As such, the saturating concentrations determined through titration should be applicable across all flow cytometers. We compared different titrations acquired in parallel in different platforms to understand if differences in the cytometer optical layout and performance (laser wavelength, laser power, filters) play a role in titer determination. Figure 9A,B show the data of anti-human and anti-mouse titrations, respectively. All instruments were used at the optimal setup based on manufacturer recommendations. In all evaluations, 18-bit files were exported for comparison using the same data transformation and scaling. 

The resolution was equivalent between instruments, with similar stain indices and optimal titer determinations. For APC/Fire 750 and APC, the highest resolution was achieved with the Aurora system due to a brighter signal and tighter background, followed by the Fortessa, and the Canto. However, the optimal Ab concentration remained unchanged, as predicted. Similar observations were found for mouse CD4 conjugated to BV421, BV785, FITC, PerCP-eFluor 710, PE, PECy7, APC, and APC/Fire 750. Burn et al. made similar observations of the optimal titer preservation when making comparisons across cytometer types [29].

We next sought to evaluate the impact of the instrument selection when titrating dimly expressed markers. We assessed human TCRγδ PE/Fire 700 (co-stained with CD3) and NKG2D Alexa Fluor 647 (co-stained with CD8). For both specificities, a similar resolution, stain index, and percentage of positive cells were observed for each Ab dilution across instruments (Figure 9C). 

### 3.7. Impact of Cell Number and Staining Volume

We evaluated whether the optimal titer selection is impacted by variation in the number of stained cells. Figure 10A shows the stain index and percentage of positives for a CD4 PE-Cy7 titration performed using different cell numbers for staining. The same optimal Ab titer was selected when staining between 50,000 and 5 × 10^6^ cells, using the same final volume (250 μL). No impactful differences in optimal titer or resolution were observed within the cell number range evaluated, as expected, since the Ab concentration, if used at saturation, would exceed the number of cellular targets. This observation is commonly applied during sorting experiments, where the number of cells to stain can be adjusted up to 10-fold, while using the same Ab concentration and with a minimal loss of resolution. 

In multicolor panels, the marker resolution can be improved by adding some reagents in sequential order, as demonstrated by Park et al. [13,22]. We evaluated whether differences in the total staining volume throughout the multi-step Ab addition process have an impact on the stain index or optimal titer for the reagents added sequentially. Figure 10B shows a titration of human TCRγδ PerCP-Vio700 at different staining volumes. A higher stain index is observed when the Ab is added at lower volume, but the final titer did not change when the staining volume ranged between 150 and 1000 μL. If a sequential staining approach is used, there are no changes in the optimal titer for reagents added in before the total volume is achieved. 

## 4. Conclusions

Ab titration is fundamental for any panel optimization and needs to be conducted before using reagents in a multicolor experiment. Every step in the titration process needs to be planned carefully, starting from sample selection down to the data analysis. Titration aims to find the concentration of Ab that best resolves a positive signal from the background. An incorrect antibody titer can lead to non-specific binding, wasted reagents, and suboptimal data resolution and quality. The lack of Ab saturation could be explained by multiple factors: a low affinity Ab, for which the optimal concentration can be exceedingly high to reach equilibrium; a diluted stock; Abs sticking to each other, causing lack of availability for binding; or signal quenching. Based on our observations, the determination of optimal antibody concentrations is necessary for each experimental paradigm, sample type, cell preparation condition, staining protocol, and reagent lot. 

We concluded that the optimal Ab titer selection was not impacted by the resolution metric, instrument setup, datafile type, AF extraction, cell numbers, staining volume, or the cytometer used. However, a higher population resolution was achieved by using a full spectral cytometer, optimized CAS acquisition settings, lower staining volume, or AF extraction. The optimal Ab concentration varies for the same marker depending on the cell preparation and staining protocols, highlighting the importance of assessing each marker separately and performing titrations under the same conditions as the ones used to stain the experimental samples. The impact of other variables on the titer selection and resolution, such as cell viability, fixation, Fc receptor blocking, incubation times, spillover spreading, still needs to be addressed.

## 5. Discussion

Multiple factors in the preanalytical and analytical steps of a flow cytometry experiment can influence the titration decision and the subsequent ability to resolve populations of interest, requiring careful experimental planning. There are five aspects to consider before starting a titration to ensure the testing will run smoothly and based on our study, we provide the following recommendations (Figure 11):

Sample quality:
Check cell viability.Ensure that enough positive cells are available for all the titration points.Use cells that represent the populations and expression levels found in the samples of study.Include both cells that express and do not express the target of interest, to assess the background contribution of cellular autofluorescence, the inherent optical and electronic noise in the instrument, and the NSB. In the NSB, the Ab binds to proteins other than the intended target, leading to inaccurate results. This can be reduced by using blocking agents or by choosing an antibody with high specificity.Reagent quality:
Review the current literature to identify the most suitable Ab clone and staining protocol for each marker of interest.Identify in advance the Ab stock concentration provided by antibody manufacturers in the certificate of analysis.Define the Ab dilutions to be evaluated, depending on the stock Ab concentration, and the expected range of concentrations to conduct the evaluation (usually between 8 to 1000 ng/test). A test is defined as the final staining volume after the reagents and cell suspension are added, usually between 250 and 300 μL for large panels.Ideally include eight to twelve serial two-fold dilutions, usually starting with twice the concentration recommended by the manufacturer (Figure 2A). Note: Optimal titers for a specific cell type/sample preparation often differ from the concentration recommended by manufacturers.
Assay design:
Determine whether you need a blocking reagent, viability dye, red blood cell lysis, platelet depletion, and/or a co-staining marker. For example, the detection of TCRγδ expression is facilitated by using co-staining with the lineage marker CD3.Verify the markers of interest can be detected under the staining conditions used in the assay, checking if they are affected by stimulation, permeabilization, or fixation methods.Set up the number of cells to stain, final staining volume, acquisition format (tube or plate), number of events to record (based on frequency of the positive population or sample availability), acquisition settings to be used, and acquisition flow rate.Verify that the flow cytometer is working optimally as per manufacturer’s specifications.Data analysis:
Before proceeding with any calculations, verify the sample and staining quality, as well as the expected fluorochrome full emission profiles, when using a spectral flow cytometer, such as the one described in this publication.Establish in advance how the titration data will be analyzed, which events will be excluded (cell debris, cell doublets, dead cells, scatter discrimination), which populations will be gated in and out, which descriptive statistics will be calculated (MFI, rSD), what metrics will be used to identify optimal titers, and what plots will be created, including a concatenated image to have a better picture of all dilutions.It is important to monitor the percentage of positive cells, to make sure the ability to capture marker positivity is preserved across the different dilutions.Of note, when using human blood samples, plotting the marker of interest against side scatter (SSC) helps to check the NSB in cells that should not express the antigen, acting as internal negative controls.The optimal titer is the concentration of the antibody that gives the best resolution, defined as the highest separation between positive and negative events and the highest percentage of positive cells identified [30]. The amount of background will depend on cellular autofluorescence, non-specific binding, reagent performance, and instrument configuration, performance, and setup. The positive signal will depend on the marker expression level, fluorochrome brightness, acquisition settings, and antibody performance and titer.Multiple metrics assess resolution, using relative values to compare across Ab dilutions and identify the concentration that provides the highest resolution. The signal to noise (SN) ratio is calculated by dividing the median fluorescence intensity (MFI) of the positive cells by the MFI of the negative cells [24]. Plotting only the MFIs is informative to assess if saturation is being reached. However, the MFI needs to be complemented to account for the spread in the negative population. The stain index is calculated as the difference between the MFI of the positive and negative populations, divided by two times the rSD of the negatives [25]. The separation index is calculated as the difference between the MFI of the positive and negative populations, divided by the ratio of the difference between the 84th and 50th percentiles of the negatives [26,27]. Both the stain index and separation index formula account for the spread in the negatives in the denominator, where the separation index gives more weight to the right side of the negative peak as a better place to look for a true resolution.For assay reproducibility, recommended Ab concentrations must be reported in ng/test or ug/mL instead of the dilution factor, since stock concentrations can vary. Expressing the titer as an amount of antibody per staining volume allows for the easy comparison of titers across reagents and reagent lots.
Panel performance:
Upon titration in single-stained cells, the Ab titer might need to be adjusted in complex multicolor assays due to differences in staining volume, competition for space when Abs are added simultaneously, or reagent interactions that cause a loss of resolution in the multicolor scenario.Adjustments could include Ab titer increase, sequential staining, or even fluorochrome substitutions. Multiple publications have described the use of single-stained cells to compare resolution against fully stained samples and identify needed changes [13,22].Titration is required for any new assay, antibody lot, clone, or conjugate. Re-titration is recommended if there are any changes in the staining protocol, lot change, storage issues (e.g., a drop in storage temperature, light exposure, reagents left out of the refrigerator by accident), fluorochrome replacements [22], or if any suspicious data are reported [24].


We have discussed single reagent titration, which is the most used approach [2,13,15]; however, other strategies have been used for optimal titer determination, including serial titrations or combinational titrations, to account for fluorochrome spillover spreading. Serial titration starts with first finding the optimal concentration for the viability dye and CD45, then titrating lineage markers in the presence of tittered CD45 and viability dye, and finally using all those conjugates as a backbone panel to titrate remaining markers, defining the optimal concentration [29,31]. Another approach is to modify the staining protocol to reduce Ab consumption. For example, Whyte C. et al. reported the use of overnight Ab staining to improve resolution while preserving specificity and reducing assay costs [32]. 

As shown, titrations are completed for each of the reagents separately. For multicolor panels, a loss of resolution can then be evaluated by comparing single-stained and fully stained cells. OMIP-069 shows titration evaluation using concatenated files, calculating the stain index, evaluating binding to monocytes, spread and brightness of the negative population, and the percentage of positive cells to determine the optimal titer for fully stained cells. Modifications in the titer or the staining protocol could be required to revert resolution loss [13]. We have also observed that a re-optimization of a previously implemented panel might reveal the need for the adjustments of titers due to differences in reagent lots [22]. The consistency in the signal intensity and population resolution and frequency needs to be evaluated every time a new reagent lot is used to assess if changes in the optimal titer are required. 

Independently of the approach used, Ab titration has proved to be a key investment in the generation of high-quality data for flow cytometry experiments, and good titration practices need to be followed to achieve accurate and reproducible results. 

## Figures and Tables

**Figure 1 cells-13-01677-f001:**
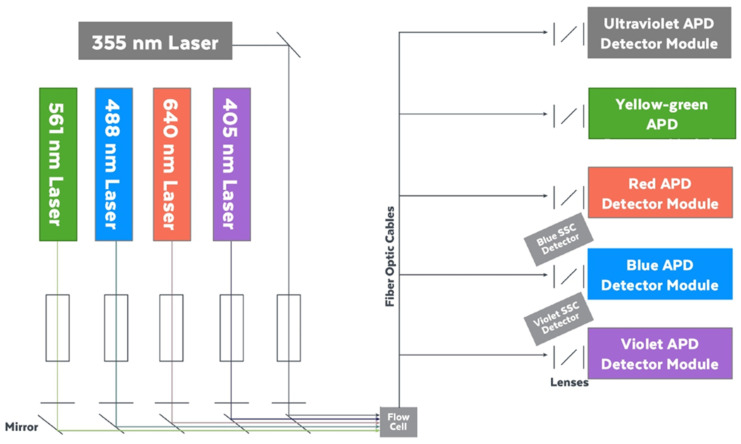
Optical Configuration Cytek Aurora Cytometer. System is equipped with five spatially separated lasers, three scatter channels, and 64 fluorescence detectors. The Aurora uses highly sensitive avalanche photodiode (APD) light detectors. Cells are interrogated by the different lasers in a single cell fashion, scatter and fluorescent light is then captured by the detectors, and finally converted into an electronic signal and visualized in plots. In the Aurora, being a spectral system, fluorescence emission is captured by every detector to build a full spectral signature per sample acquired.

**Figure 2 cells-13-01677-f002:**
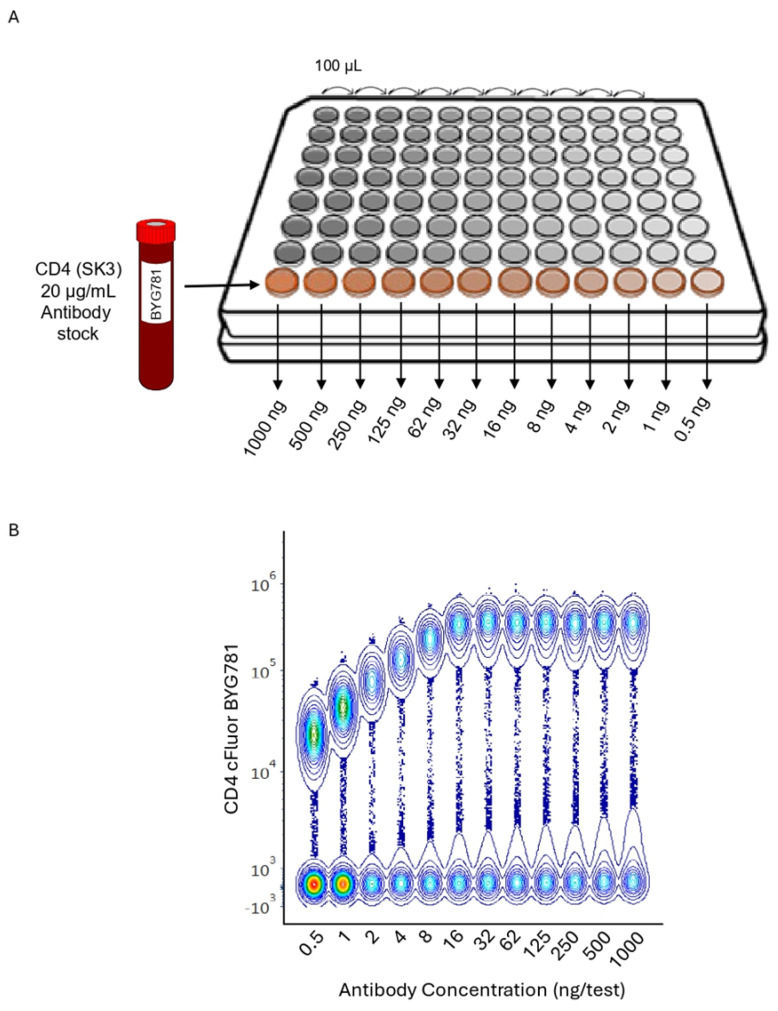
Preparation of a CD4 human PBMC Titration and Data Analysis. (**A**) How to perform a twelve 2-fold serial antibody dilution using per tube 200,000 PBMCs stained with anti-human CD4 cFluor® BYG781 (Clone SK3) starting with a stock of 20 μg/mL and (**B**). concatenated files to evaluate the titration progression across concentrations. (**C**). The graph presents the concentration of antibody used (expressed in ng per test) versus the stain index. Representative pseudo color plots gated on lymphocytes displaying SSC vs. CD4 are shown for each section. The doted line indicates optimal saturating Ab concentration.

**Figure 3 cells-13-01677-f003:**
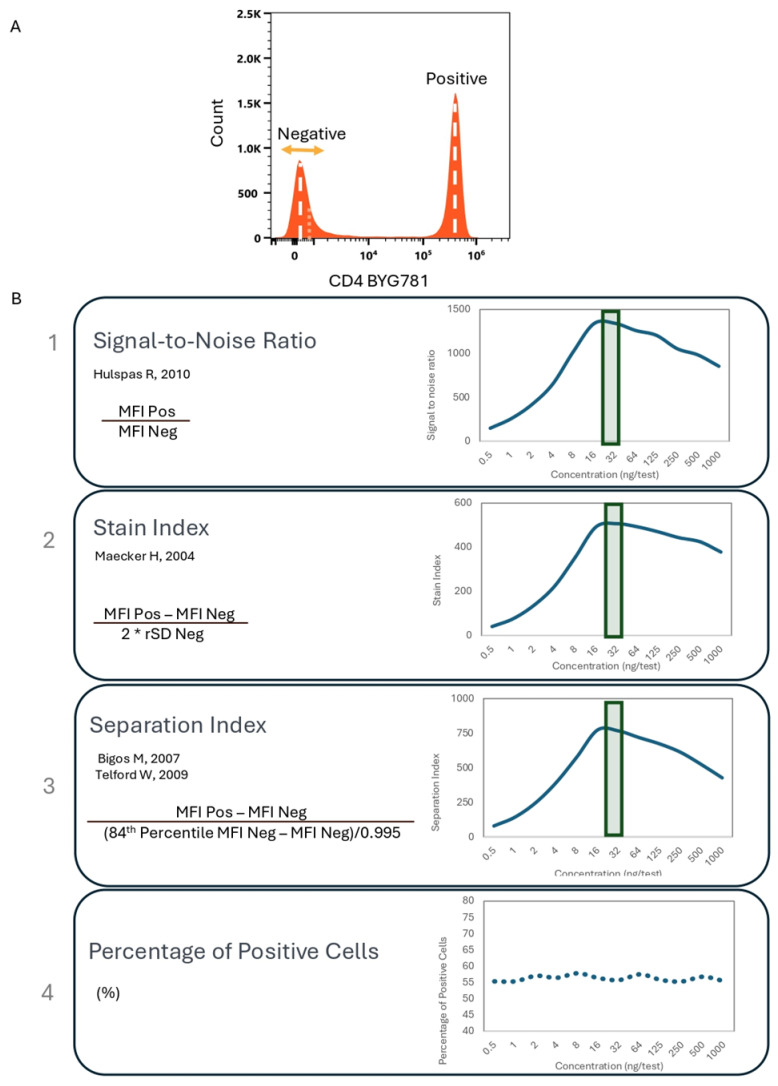
Metrics Used to Define Optimal Titer. (**A**) Identifying the optimal separation between positive and negative populations can be done using different metrics (**B**) [24,25,26,27], all of them resulting in similar patterns. MFI = Median Fluorescent Intensity, Pos = Positive population, Neg = Negative population, rSD = robust standard deviation. Optimal Ab concentration indicated by a green rectangle.

**Figure 4 cells-13-01677-f004:**
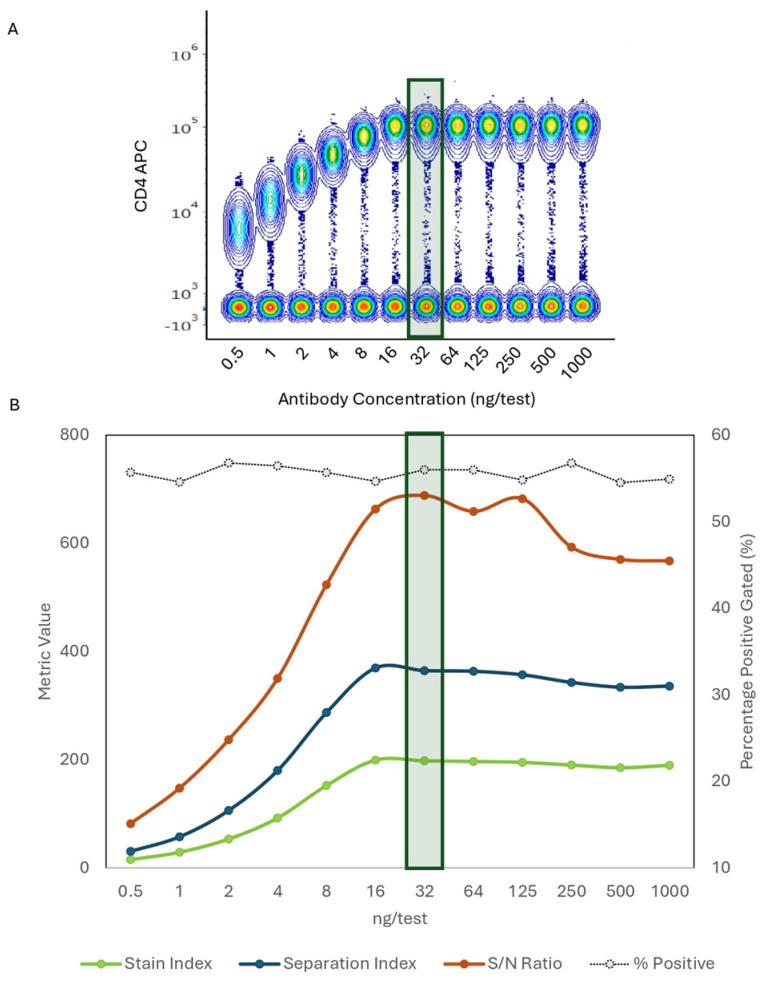
Population Resolution and Optimal Titer Comparison Across Metrics. Different metrics to assess optimal titers produced similar results. (**A**). Concatenated data from twelve 2-fold dilutions of PBMCs stained with anti-human CD4-APC (Clone SK3). At 32 ng/test, antibody saturation was reached. (**B**). The Stain Index, Separation Index, signal-to-noise (S/N) ratio, and percentage (%) of positive cells from anti-human CD4-APC were calculated, producing comparable titration curves resulting in the same optimal titer selection. Diagram represents the ng per test used (x axis), the value obtained for each metric (y axis) and the percentage of positive cells (second y axis). (**C**). Concatenated data from twelve 2-fold dilutions using different antibody conjugates (PE/Cy7, PE) but with the same clone (Clone SK3). The Stain Index, Separation Index, Signal-to-noise (S/N) ratio, and percentage (%) of positive cells were calculated, resulting in similar curve patterns and optimal titer. (**D**). Concatenated data from seven 2-fold dilutions of PBMCs stained with anti-human TCRγδ-PerCP-Vio700, showing the optimal titer at 500 ng/test. (**E**). Graph shows how all metrics produced comparable curves resulting in the same optimal titer, whereas the % of positive cells increased as the antibody concentration increased. Diagram represents the ng/test used (x axis), the value obtained for each metric (y axis) and the percentage of positive cells (secondary y axis). The green rectangles indicate optimal Ab concentration.

**Figure 5 cells-13-01677-f005:**
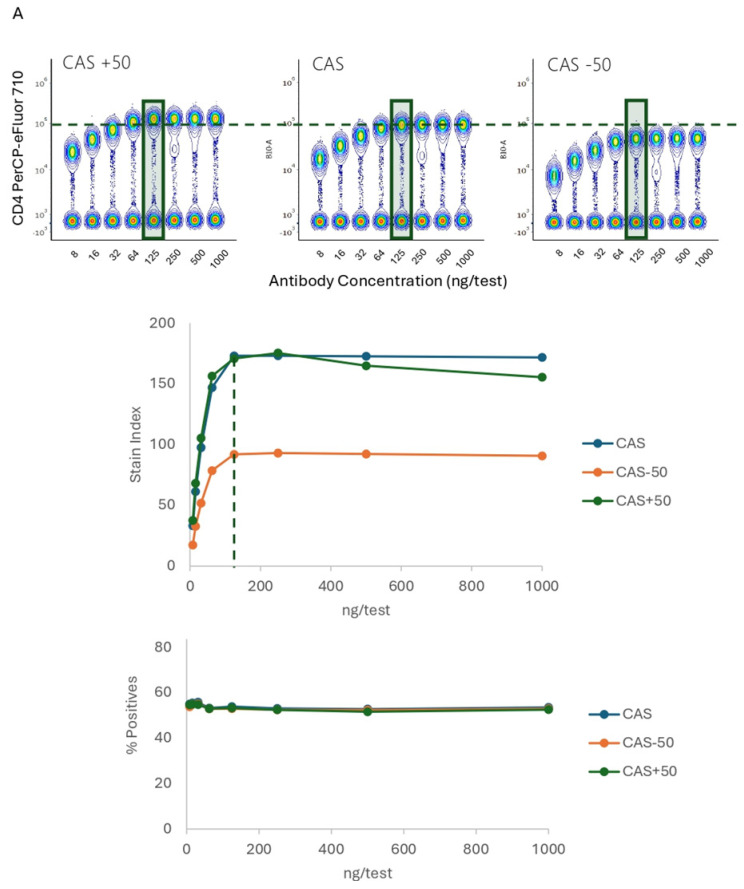
Resolution and Optimal Titer Comparison across Instrument Settings. (**A**). Concatenated data from seven 2-fold dilutions of PBMCs stained with anti-human CD4 PerCP-eFluor710, stain index and percentage of positive events are shown. The same samples were acquired at different settings, including the ones recommended by the cytometer manufacturer, named CytekAssaySetting (CAS) and values 50% below or over the recommended settings. (**B**). Concatenated data from PBMCs stained with anti-human CD314 BUV615, stain index and percentage of positive events are shown. The same samples were acquired at different settings. (**C**). Concatenated data from PBMCs stained with anti-human TCRγδ-PerCP-Vio700, stain index and percentage of positive events are shown. The same samples were acquired at different settings. CD314 BUV615 was costained with CD8 BB515 and TCRgd PE/Fire700 with CD3 BV570, to facilitate the identification of positive events. The green rectangles and dotted lines indicate optimal Ab concentration.

**Figure 6 cells-13-01677-f006:**
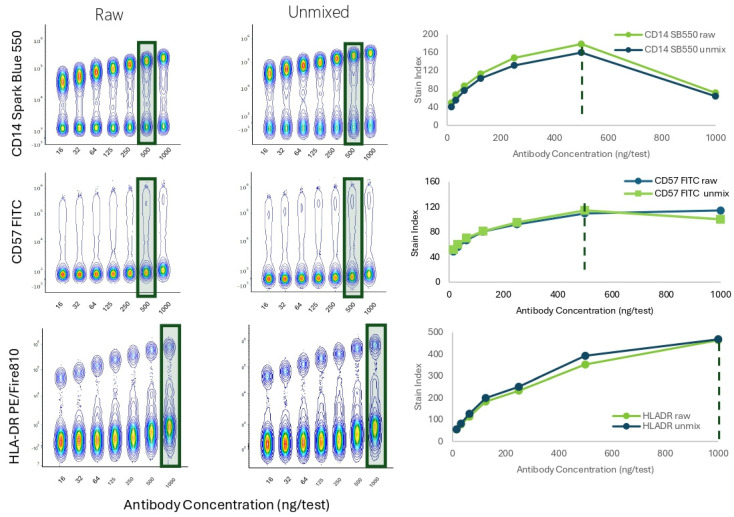
Population Resolution and Optimal Titer Comparison Across Data File Types. PBMCs were stained with anti-human CD14 SparkBlue550, anti-human CD57 FITC, or anti-human HLA-DR PE/Fire810. Concatenated data from seven 2-fold dilutions for each marker is presented. Raw data was obtained using the peak channel of each molecule as shown on the Full Spectrum Viewer of the Cytek Cloud (Peak channels: SparkBlue 550 = B3, FITC = B2, PE/Fire810 = YG10). The Stain Index was calculated using raw or unmixed data, resulting is comparable titration curves and optimal titers. The green rectangles and doted lines indicate optimal Ab concentration.

**Figure 7 cells-13-01677-f007:**
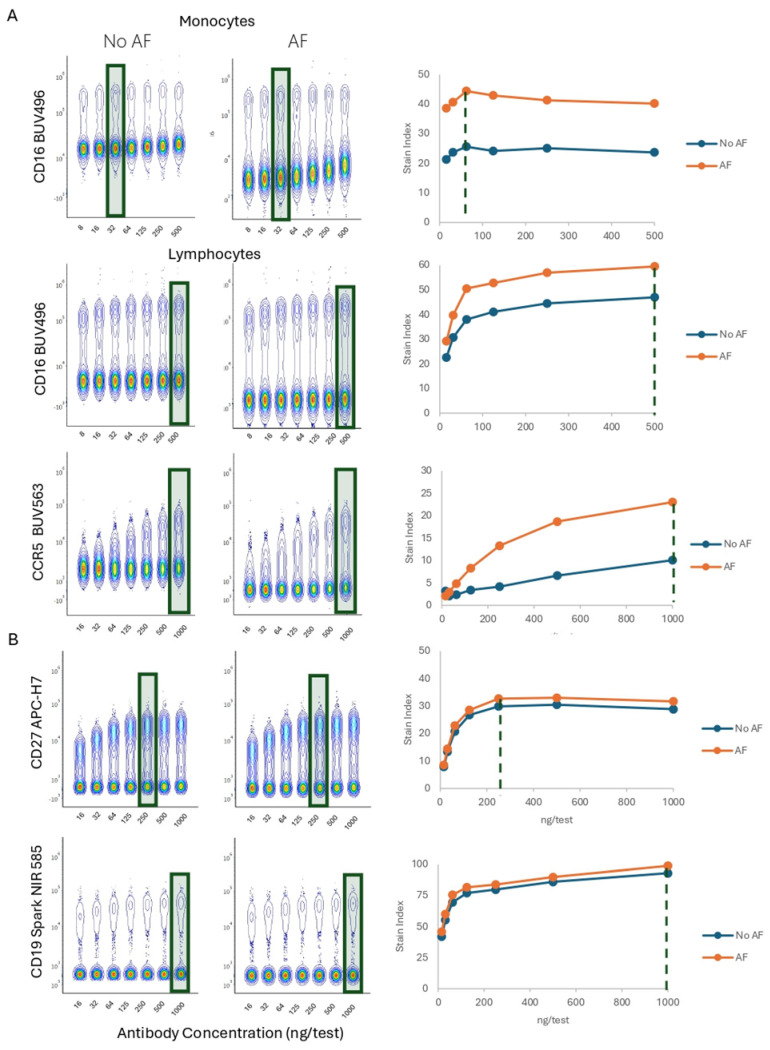
Population Resolution and Optimal Titer Comparison Across AF management strategies (**A**,**B**). PBMCs were stained with anti-human CD16 BUV496 or anti-human CCR5 BUV563. CD16 was unmixed without (No AF) or with AF extraction (AF), either using monocytes or lymphocytes to define autofluorescence. Concatenated data and titration curves are presented. The green rectangles and doted lines indicate optimal Ab concentration.

**Figure 8 cells-13-01677-f008:**
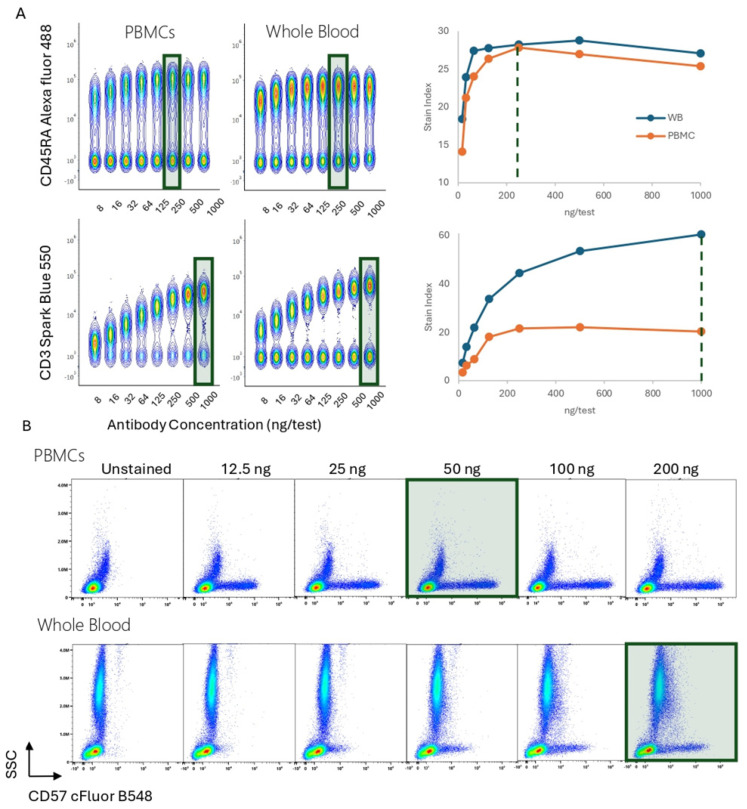
Population Resolution and Optimal Titer Comparison Across Sample Preparation Conditions. (**A**). PBMCs or whole blood were stained with anti-human CD45RA Alexa Fluor 488 or anti-human CD3 Spark Blue 550. Concatenated data is presented when using PBMCs or whole blood. Stain index for each marker and condition were calculated. Titration curves are shown. (**B**). CD57 cFluor B548 pseudo color plots shown for PBMCs and whole blood. Concentrations from 12.5 to 200 ng were tested. The green rectangles and doted lines indicate optimal Ab concentration.

**Figure 9 cells-13-01677-f009:**
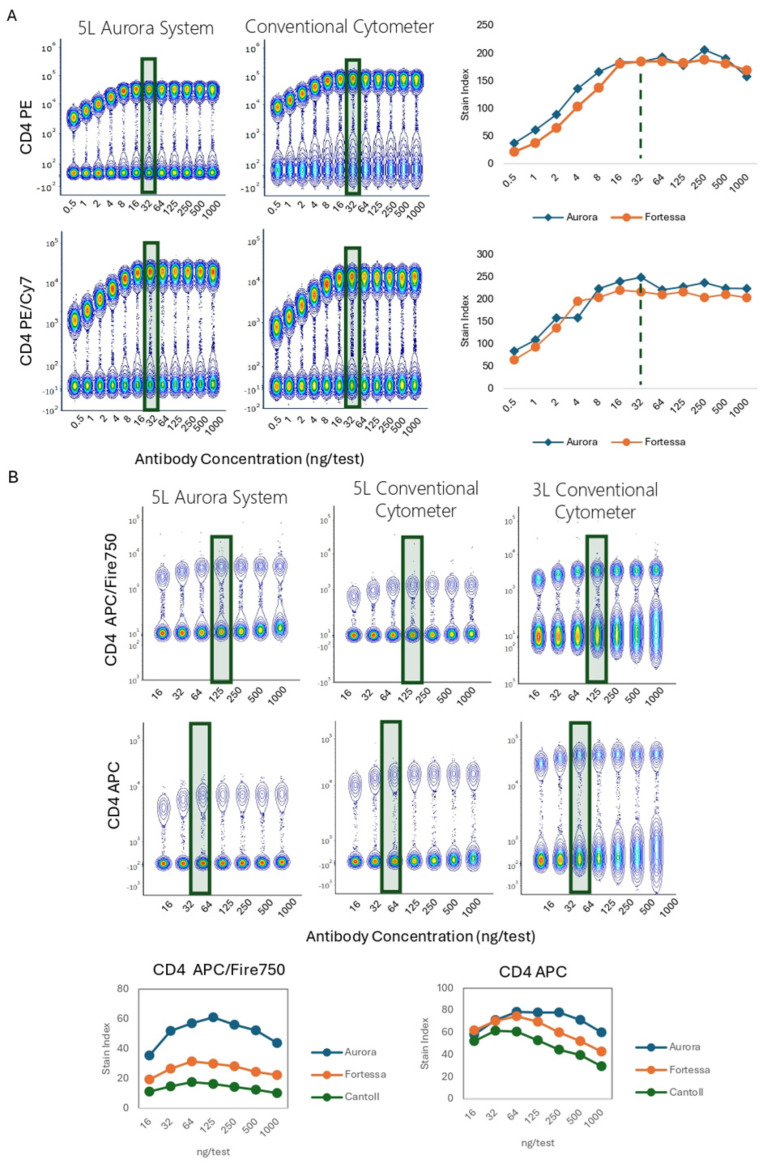
Population Resolution and Optimal Titer Comparison Across Flow Cytometers. (**A**). PBMCs were stained with anti-human CD4 PE, or anti-human CD4 PE/Cy7, and acquired in a Cytek Aurora 5 L or a conventional instrument equipped with 4 L. Concatenated data is presented. Stain indexes were calculated. (**B**). Mouse splenocytes were stained with anti-mouse CD4 APC/Fire 750, or anti-mouse CD4 APC, and acquired in a Cytek Aurora 5 L or a conventional instrument equipped with 5 or 3 L. (**C**). Human PBMCs were stained with anti-human TCRγδPE/Fire700 or anti-human NKG2D Alexa Fluor 647 and acquired in a Cytek Aurora 5 L or a conventional instrument equipped with 4 L. Titration curves and optimal titers are shown. TCRγδPE/Fire700 was co-stained with CD3 and NKG2D with CD8, to facilitate the identification of positive events. The green rectangles and doted lines indicate optimal Ab concentration. (L = laser).

**Figure 10 cells-13-01677-f010:**
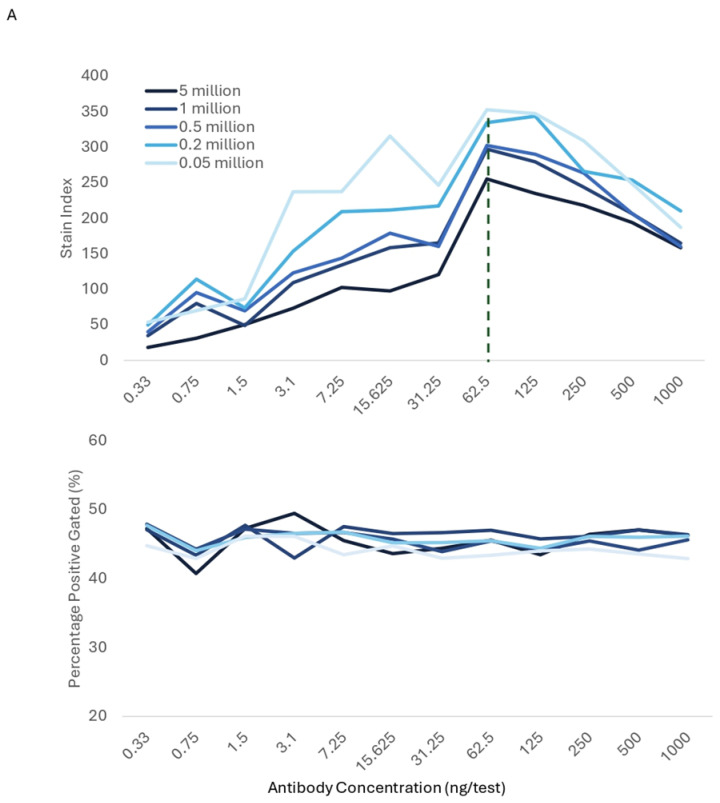
Population Resolution and Optimal Titer Comparison Across Cell Numbers and Staining Volumes. (**A**). PBMCs were stained with anti-human CD4 PE/Cy7, using different cell numbers for staining, ranging between 50,000 and 5 × 106 cells. (**B**). PBMCs were stained with anti-human TCRγδ PerCP-Vio700, using different staining volumes, ranging from 150 to 1000 μL. Stain indexes and percentage of positive cells were calculated. The dotted line indicates optimal Ab concentration.

**Figure 11 cells-13-01677-f011:**
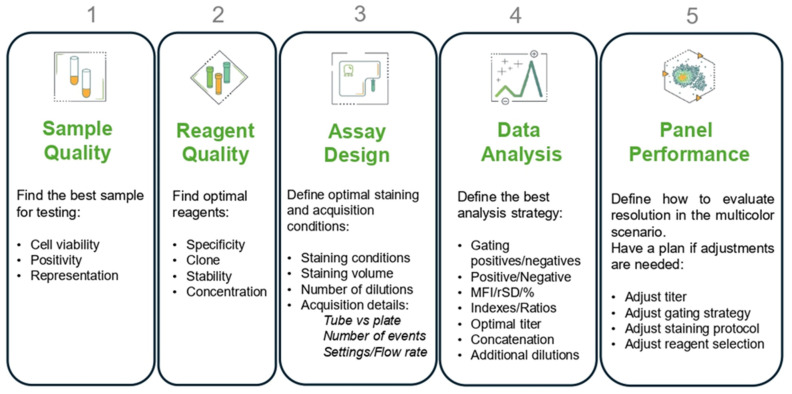
Technical Aspects to Consider When Titrating Reagents.

## Data Availability

In the submitted manuscript, we have share protocols, processed data, and the list of relevant publications. The flow cytometry fcs files used in this publication are considered Cytek proprietary data, but they will be available to the readership or the reviewers upon request from the corresponding author.

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
