# Peer review of "The Power of Reagent Titration in Flow Cytometry"

_cells, 2024, doi:10.3390/cells13201677_

Round 1
Reviewer 1 Report
Comments and Suggestions for Authors
This manuscript includes lots of survey facts and analyses, which may give good guidance to a wide range of readers. From this positive view point, I may suggest publications of this work in Cells. However, improvements of understandability would be necessary upon revision to give more impacts to the research communities. Please see below.
1) It is rather difficult to expect the paper contents from this paper title. It is too simple. It is better to add a suitable subtitle.
2) Addition of one illustration for general outline structure of flow cytometry system is recommended for better understanding of Figure 1.
3) Section name always start from 1. Something is wrong. please fix it.
4) Addition of one figure for summary of discussion part is helpful.
Author Response
- This manuscript includes lots of survey facts and analyses, which may give good guidance to a wide range of readers. From this positive viewpoint, I may suggest publications of this work in Cells. However, improvements of understandability would be necessary upon revision to give more impacts to the research communities. Please see below.
Thank you for reading and reviewing our manuscript. We are grateful for the feedback to improve the quality of the publication. We incorporated all of the suggestions.
- It is rather difficult to expect the paper contents from this paper title. It is too simple. It is better to add a suitable subtitle.
Thank you for pointing this out, we agree with this comment. As suggested, we have included the subtitle “Impact of Sample Preparation, Staining Protocol and Instrumentation Features in Resolution and Titration Performance” to accompany the original title.
- Addition of one illustration for general outline structure of flow cytometry system is recommended for better understanding of Figure 1.
Agree. As suggested, a new figure has been included to explain the configuration of a cytometer. Figure 1 has been included, and figure numbers and citations have been updated accordingly.
- Section name always start from 1. Something is wrong. please fix it.
We apologize for this issue. The numbering was correct in the provided word document, unfortunately something happened during the file conversion on the submission page. We will confirm that no such issue is observed again.
- Addition of one figure for summary of discussion part is helpful.
Agree. Figure one has been moved to the end as a summary figure, as well as the related content to support the newly included discussion session, suggested by another reviewer.
Reviewer 2 Report
Comments and Suggestions for Authors
This is a well written and excellent article on the utility of antibody titration. It will be of great use for researchers who are setting up flow cytometry immunofluorescence experiments. Unfortunately however, the formatting of the figures within the review article pdf is not very good, and Figures 7-10 are missing (eg Figure 7 referred to in the text Line 464-465). This needs to be fixed, and the corrected manuscript sent out again for review before it can be approved for publication.
There are a few minor errors that were found as well:
Abstract Line 26 - ‘NBS’ should be ‘NSB’
The numbering of the sections after Figure 1 appears wrong (eg Line 124 Reagent Quality is listed as ‘1.’ when it is 2 in Figure 1)
Line 120 - ‘NBS’ should be ‘NSB’
Line 222 – ‘(orange section)’ is actually green in Figure 2. Need to change graph coloring or text.
Again, in Result section there are multiple sub-sections that are numbered ‘1.’ This should be corrected.
Author Response
Comment1: This is a well written and excellent article on the utility of antibody titration. It will be of great use for researchers who are setting up flow cytometry immunofluorescence experiments. Unfortunately, however, the formatting of the figures within the review article pdf is not very good, and Figures 7-10 are missing (eg Figure 7 referred to in the text Line 464-465). This needs to be fixed, and the corrected manuscript sent out again for review before it can be approved for publication.
Response 1: Thank you for your feedback and for helping us to fine-tune the content of the manuscript. Figure order and formatting has been reviewed and corrected.
Comment2: There are a few minor errors that were found as well: Abstract Line 26 - ‘NBS’ should be ‘NSB’
Response 2: Error has been fixed accordingly.
Comment 3: The numbering of the sections after Figure 1 appears wrong (eg Line 124 Reagent Quality is listed as ‘1.’ when it is 2 in Figure 1)
Response 3: Order of the sections has been revised and corrected.
Comment 4: Line 120 - ‘NBS’ should be ‘NSB’
Response 4: Error has been fixed accordingly.
Comment 5: Line 222 – ‘(orange section)’ is actually green in Figure 2. Need to change graph coloring or text.
Response 5: Thank you for pointing this out, the text has been modified to cite the actual green color.
Comment 6: Again, in Result section there are multiple sub-sections that are numbered ‘1.’ This should be corrected.
Response 6: We apologize for this issue. The numbering was correct in the provided word document, unfortunately something happened during the file conversion on the submission page. We will review that no such issue is observed again.
Reviewer 3 Report
Comments and Suggestions for Authors
Authors summarized the fundamentals of titration best practices, and evaluated the impact of using different samples, staining protocols, acquisition settings, and analysis conditions in the selection of optimal titer and population resolution. Although the content is interesting and well presented, the article lacks of a solid format including introduction, methods, results, discussion, limitations and conclusion. Authors should reformat their manuscript with appropriate headings and subheadings, making it more appropriate for readers.
Figures are well-illustrated. However, it would be better if you tried to merge some figures in one.
Author Response
Comment 1: Authors summarized the fundamentals of titration best practices, and evaluated the impact of using different samples, staining protocols, acquisition settings, and analysis conditions in the selection of optimal titer and population resolution.
Response 1: Thank you for reading and reviewing our manuscript. We are grateful for the feedback to improve the quality of the publication. We incorporated all of the suggestions.
Comment 2: Although the content is interesting and well presented, the article lacks of a solid format including introduction, methods, results, discussion, limitations and conclusion. Authors should reformat their manuscript with appropriate headings and subheadings, making it more appropriate for readers.
Response 2: Thank you for pointing this out, we agree with this comment. Initially, the manuscript was intended to be a review article, but the addition of a result section made it fit into a different category. Manuscript has been reformatted to follow reviewer’s recommendations.
Comment 3: Figures are well-illustrated. However, it would be better if you tried to merge some figures in one.
Response 3: Figures 10 and 11 were merged into 1, as well as the sections referencing them. The information in those figures is now included in Figure 11.